# TUNE: FREQUENCY-GUIDED TOKEN GATING FOR ROBUST CONTINUAL LEARNING IN LLMS

## ABSTRACT

Continual learning (CL) in large language models (LLMs) remains a critical challenge, as sequential training often results in catastrophic forgetting of previously learned knowledge. To our knowledge, no prior work has approached CL in LLMs from a frequency perspective, despite strong evidence that spectral properties directly govern model robustness and vulnerability to forgetting. Recent methods based on Low-Rank Adaptation (LoRA) have shown promise for parameter-efficient CL, but remain preliminary, relying on task-specific subspace expansion with additional regularization. We propose TUNE (Token Update via Noise-robust Frequency Encoding), a frequency-guided token modulation mechanism that stabilizes LoRA residual updates. TUNE employs a stationary wavelet transform (SWT) to decompose token embeddings into multi-resolution subbands, where token saliency is derived from high-frequency activations and reliability is assessed through cross-scale agreement. These signals are fused into token-wise scaling that amplify reliable updates while suppressing noisy fluctuations. Without introducing additional trainable parameters beyond LoRA expansion, TUNE achieves significant improvements over the SOTA baselines, establishing frequency-aware token adaptation as a promising direction for CL in LLMs.

## 1 INTRODUCTION

**Continual Learning in LLMs**  Continual learning (CL) aims to adapt models to new tasks while retaining prior knowledge (Silver et al., 2013; De Lange et al., 2021). This is especially critical for Large language models (LLMs), which operate in evolving domains at scale. Yet, when fine-tuned or continually trained on new data, LLMs often suffer from catastrophic forgetting—the tendency to overwrite previously acquired knowledge (Luo et al., 2023; Shi et al., 2024; Li et al., 2024). This phenomenon limits the adaptability and reliability of LLMs, making it a critical area of research. Existing CL paradigms include (i) rehearsal Huang et al. (2024); Pillai (2025), (ii) regularization such as EWC/SI Kirkpatrick et al. (2017); Zenke et al. (2017), and and (iii) architecture-based subspace isolation or expansion strategies Mallya & Lazebnik (2018); Wang et al. (2023b). While effective, these methods are often computationally overload for LLMs due to their massive scale.

Recent advances in parameter-efficient fine-tuning (PEFT) have shifted attention toward lightweight adaptation methods. In particular, Low-Rank Adaptation (LoRA) Hu et al. (2021) inserts small trainable low-rank matrices while freezing the original weights, enabling efficient continual updates. Extensions such as O-LoRA Wang et al. (2023a) and N-LoRA Yang et al. (2024) introduce explicit regularization on inter-task orthogonality or parameter collision to mitigate forgetting while preserving efficiency. These approaches highlight the promise of PEFT-based CL, where the cost of adaptation is significantly reduced.

**High-Frequency Drift as the Substrate of Forgetting**  Prior work has shown that model robustness is closely tied to its frequency response (Yin et al., 2020; Ilyas et al., 2019; Sun et al., 2024). Spectral analyses consistently reveal that low-frequency components encode the core global context, serving as the primary carrier of semantics that discriminates between tasks or domains. In contrast, high-frequency components act largely as detail enhancers—capturing local, instance-specific refinements rather than general task-level structure (Cooley et al., 1969; Rahaman et al., 2019; Zhi-Qin John Xu et al., 2020; Pan et al., 2023). Since they emphasize sample-wise variations, high-frequency features are typically noisier and less stable, making them unreliable anchors for continual task adaptation. Such abrupt feature drift in high-frequency band causes unstable representation shifts across tasks and makes it a primary substrate of catastrophic forgetting.

To our knowledge, no prior work has addressed continual learning in LLMs from a frequency perspective. In spectral view, low-frequency components provide the stable semantic backbone for each task, while high-frequency components capture instance-wise local variations that are useful for capturing details but also prone to drift. This contrast is evident in Figure 1: low-frequency coefficients ($A_{10}$) form well-separated clusters between tasks, reflecting their role as stable semantic carriers, whereas high-frequency coefficients ($D_1$) collapse into overlapping clusters scattered broadly, showing their noisy property.

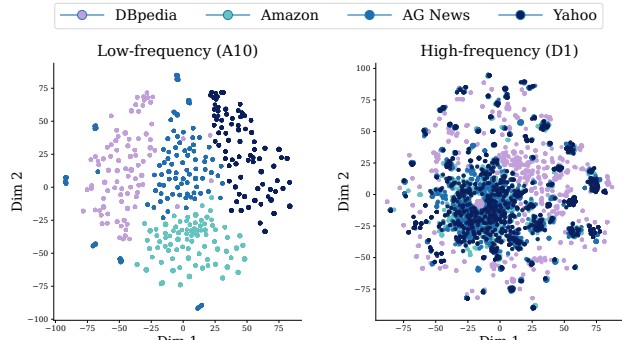

Figure 1: **t-SNE visualization of SWT-decomposed token features across 4 different tasks in standard CL Benchmark.** Left: level-10 low-frequency coefficients ($A_{10}$) exhibit clear separation by task. Right: level-1 high-frequency coefficients ($D_1$) show overlapping clusters.

Motivated by the high-frequency-induced challenges in continual learning, we propose **TUNE** (Token Update via Noise-robust Frequency Encoding), a spectral token gating mechanism that integrates with LoRA residual adaptation to mitigate noisy high-frequency drift while amplifying reliable, salient signals. TUNE implements this by applying a stationary wavelet transform (SWT) along each token embedding sequence, decomposing it into aligned multi-resolution frequency subbands while strictly preserving original token structure. This decomposition enables multi-scale inspection of token activity and identifies whether the given token features are stable or noise-prone.

Specifically, TUNE introduces two complementary measures. First, a high-frequency saliency score quantifies the strength of token-level fine-grained variation, highlighting where meaningful novelty arises. Second, a parent-guided reliability score measures the consistency of high-frequency activations across scales, i.e., the agreement between a child detail coefficient and its reliable parent in the SWT hierarchy. This agreement-based reliability suppresses unstable spikes that fail to persist across scales while emphasizing coefficients with strong cross-scale support. Combining these two measures yields a token-dependent gating factor that rescales the LoRA residual updates, assigning larger adaptation capacity to reliable, salient tokens while damping noisy ones. In doing so, TUNE curbs catastrophic forgetting mainly caused by high-frequency drifts, while directing token plasticity toward trustworthy novel signals. We summarize our main contributions as follows:

- To our knowledge, this is the first work to tackle continual learning in LLMs explicitly from a frequency perspective, identifying high-frequency drift as a primary substrate of catastrophic forgetting.

- We propose TUNE, a spectral token gating mechanism that integrates SWT with LoRA residual adaptation. TUNE combines high-frequency saliency with parent-guided reliability to produce token-dependent gating factors that regulate adaptation.

- TUNE introduces no additional trainable parameters beyond standard continual LoRA paradigm with minimal computational overhead.

- Extensive experiments on both small- and large-scale CL benchmarks demonstrate that TUNE consistently improves over strong state-of-the-art baselines.

## 2 RELATED WORKS

### 2.1 LoRA IN CONTINUAL LEARNING

Low-Rank Adaptation (LoRA) (Hu et al., 2021) constrains task adaptation to a low-rank update $\Delta W = AB$, enabling parameter-efficient finetuning by freezing the pretrained weights. This approach has recently been extended to continual learning settings. In particular, Orthogonal LoRA (O-LoRA) (Wang et al., 2023a) allocates a new LoRA subspace for each task while freezing those from previous tasks, and further imposes an orthogonality constraint so that the update directions of new tasks do not overlap with earlier ones. LB-CL (Qiao & Mahdavi, 2024) further develops this line by balancing LoRA consolidation across tasks to improve stability. Non-collision LoRA

(N-LoRA) (Yang et al., 2024) instead focuses on preventing parameter collisions by enforcing sparsity on task-specific LoRA parameters, ensuring that updates from different tasks occupy distinct, non-overlapping coordinates. These approaches establish LoRA as a representative backbone for continual learning in LLMs.

## 2.2 FREQUENCY PERSPECTIVE ON CATASTROPHIC FORGETTING

Representation drift caused by noisy features has been identified as a major driver of catastrophic forgetting. When models are updated on new tasks, parameters often overfit to unstable or spurious patterns in the incoming data, producing abrupt representational change that overwrites consolidated knowledge from earlier tasks. Prior work has shown that catastrophic forgetting is tightly linked to such drift. Ramasesh et al. (2020) dissected layer-wise representation dynamics and found that forgetting coincides with sharp shifts in internal activations. Caccia et al. (2022) further demonstrated that sudden representational change across tasks is predictive of performance collapse. Toneva et al. (2019) revealed that examples most frequently forgotten are those aligned with noisy or unstable decision boundaries. Together, these studies suggest that catastrophic forgetting emerges from updates driven by noisy, non-robust features that destabilize the feature space across tasks.

Building on this view, recent analyses suggest that the noisy features most responsible for drift often reside in high-frequency components of the representation. While low-frequency bands encode stable, task-relevant structure, high-frequency bands tend to amplify local variations and spurious correlations that carry weak or noisy semantics (Cooley et al., 1969; Ilyas et al., 2019; Yin et al., 2020; Pan et al., 2023; Sun et al., 2024). There is converging evidence that high-frequency components are the primary channel for model brittleness. Ablation studies conducted in Abello et al. (2021) shows that the largest error sensitivity is placed in mid/high bands and Wang et al. (2020) demonstrates that small-norm adversarial attacks concentrate their perturbations at mid/high frequencies while robust models tend to shift attribution toward low-frequency cues. These findings support treating high-frequency activations as unstable carriers of non-robust features that are prone to triggering abrupt shifts when emphasized during training. In continual learning setting, this implies that high-frequency activations can serve as a marker of noisy or fragile examples that destabilize previously learned representations if not treated carefully.

## 2.3 STATIONARY WAVELET TRANSFORM (SWT)

**Discrete Wavelet Transform (DWT)**    The Fourier transform decomposes a sequence into sinusoidal bases, fully describing its frequency content but without temporal localization, since each coefficient reflects contributions from the entire signal. This limitation makes it less suitable for non-stationary data such as language, where preserving the temporal structure is essential. The wavelet transform overcomes this by analyzing a signal $x(t)$ with functions localized in both time and frequency. These functions are generated by shifting and scaling a prototype function, the mother wavelet $\psi$. This is particularly effective for signals whose statistical properties vary over time, such as language sequences where both global and local dependencies must be captured.

Since our focus is on language data, which naturally appears as discrete token sequences, we restrict our attention to the discrete formulation of the transform. For discrete signals, this expansion is realized through the discrete wavelet transform (DWT), implemented via a pair of analysis filters for low-pass ($h$) and high-pass ($g$) decomposition, together with corresponding synthesis filters $\tilde{h}, \tilde{g}$ for reconstruction. Writing convolution as $(f * k)[n] = \sum_m f[m]\, k[n-m]$, the level-$j$ DWT coefficients and its reconstruction are computed as

$$A_j[n] = (A_{j-1} * h)[2n], \qquad D_j[n] = (A_{j-1} * g)[2n], \qquad A_0 \equiv x,$$

$$A_{j-1}[n] = (\uparrow 2\, A_j * \tilde{h})[n] + (\uparrow 2\, D_j * \tilde{g})[n],$$

where $\uparrow 2$ denotes upsampling operator by two. The recursive filtering and decimation imply that each scale isolates a specific portion of the frequency spectrum. In the dyadic DWT with orthonormal, ideal half-band filters, the frequency support of the coefficients is exactly partitioned as

$$\Omega(A_j) = \left\{ \omega : \ |\omega| \leq \tfrac{\pi}{2^j} \right\}, \qquad \Omega(D_j) = \left\{ \omega : \ \tfrac{\pi}{2^j} < |\omega| \leq \tfrac{\pi}{2^{j-1}} \right\}, \quad \omega \in [-\pi, \pi).$$

For practical wavelets such as Symlets and Daubechies, the separation between bands is only approximate. Yet $A_j$ consistently captures progressively coarser low-frequency trends, while $D_j$ emphasizes higher-frequency, localized variations.

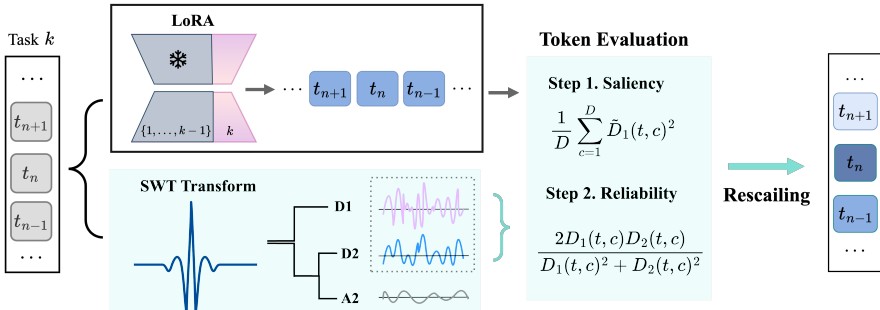

Figure 2: **Overview of TUNE.** Sequence of token embeddings are decomposed by SWT into three subbands ($A_2$: low-frequency approximation, $D_2$: mid-frequency detail, $D_1$: high-frequency detail). Then evaluate for saliency and reliability based on $D_1$ and $D_2$ coefficients, producing token-dependent scaling factors that modulate LoRA residual updates for current task $k$.

**Stationary Wavelet Transform (SWT)**  While the DWT provides an efficient multi-resolution decomposition, its use of decimation introduces shift variance, a small translation of the input sequence can cause large changes in the wavelet coefficients. This sensitivity is undesirable for language data, where precise token ordering is important to capture semantics. The stationary wavelet transform (SWT), also called the undecimated or shift-invariant wavelet transform, removes the downsampling step and instead upsamples the filters at each level. As a result, all subbands retain the same length as the original signal, and coefficient alignment across scales is preserved.

Formally, let $h^{(j)}$ and $g^{(j)}$ denote the level-$j$ analysis filters obtained by inserting $2^{j-1} - 1$ zeros between each of the filter coefficients of $h$ and $g$. The recursive SWT decomposition and reconstruction are computed as

$$A_j[n] = (A_{j-1} * h^{(j)})[n], \qquad D_j[n] = (A_{j-1} * g^{(j)})[n], \qquad A_0 \equiv x,$$

$$A_{j-1}[n] = (A_j * \tilde{h}^{(j)})[n] + (D_j * \tilde{g}^{(j)})[n].$$

After $j$ levels of decomposition, the SWT produces a set of coefficient components

$$\{A_j, D_j, D_{j-1}, \ldots, D_1\},$$

where $A_j$ represents the coarsest approximation (lowest-frequency trend) and $\{D_\ell\}_{\ell=1}^{j}$ capture progressively finer detail bands ordered from low to high frequency. Unlike the DWT, no decimation is applied, so each $A_j$ and $D_j$ has the same length as $x$. While this redundancy makes SWT less efficient, it ensures shift invariance and exact token-level alignment across scales. As a result, SWT offers a decomposition that is both semantically stable and structurally faithful to discrete token sequences.

**Interscale Correlation in SWT**  A distinctive property of wavelet coefficients is their persistence across scales: significant coefficients tend to propagate from child $D_j[n]$ to parent $D_{j+1}[n]$, while noise coefficients do not (Xu et al., 1994; Luisier et al., 2006; He et al., 2015). This phenomenon, often referred to as the parent–child relationship, reflects strong interscale correlation, where large (small) child coefficients are likely to correspond to large (small) parent coefficients. Xu et al. (1994) proposed to quantify this propagation by a normalized correlation index measuring the raw coefficient correlation between child $W_{j,n}$ and parent $W_{j+1,n}$ at location $n$. Luisier et al. (2006) further extend this property to develop inter-scale-dependent thresholding functions where weak parent-child correlation enforces stronger attenuation to suppress noise whereas high inter-scale consistency preserves reliable signal structure. By exploiting this inter-scale correlation as a key indicator of feature reliability in SWT, one can identify coefficients that carry robust linguistic information while suppressing unstable, noise-driven activations.

## 3 METHOD

We now describe Token Update via Noise-robust Frequency Encoding (TUNE), our frequency-aware mechanism for regulating LoRA residual adaptation. The key idea is to decompose token embeddings

into multi-resolution subbands and construct a token-dependent gating factor that amplifies reliable novelty while suppressing noisy, unstable fluctuations. This section details how TUNE leverages stationary wavelet transform (SWT) to extract frequency-specific signals and integrates them into the LoRA update path to mitigate catastrophic forgetting. See Figure 2 for full diagram.

## 3.1 SETTING AND NOTATION

Let $x \in \mathbb{R}^{L \times D}$ denote a token sequence with length $L$ and hidden size $D$. Applying a two–level stationary wavelet transform (SWT) with analysis filters $(h, g)$ produces three subbands

$$A_2, \ D_2, \ D_1 \in \mathbb{R}^{L \times D},$$

where $A_2$ is the level-2 low-pass approximation, $D_2$ the level-2 coarser detail, and $D_1$ the level-1 finest detail. For practical finite impulse response (FIR) filters such as Daubechies and Symlets, the frequency separation across bands is approximate, but the dominant ordering, high: $D_1 >$ mid: $D_2 >$ low: $A_2$, consistently holds.

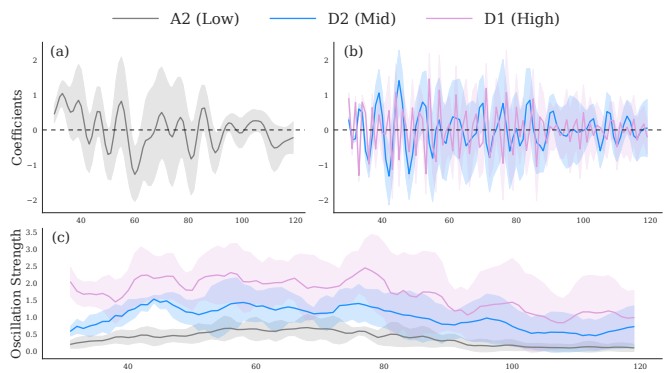

Following the inter-scale correlation framework (Xu et al., 1994; Luisier et al., 2006; He et al., 2015), we treat $D_1$ as the child and $D_2$ as its parent in the hierarchy, while $A_2$ provides the low-frequency semantic backbone. Because SWT is undecimated, all subbands retain length $L$, ensuring exact positional alignment between coefficients across scales. We index tokens by $t \in \{1, \ldots, L\}$ and channels by $c \in \{1, \ldots, D\}$.

Figure 3 illustrates this frequency hierarchy on 5 example token sequences where $A_2$ varies smoothly and forms a stable context, whereas $D_2$ and especially $D_1$ display stronger oscillations. Figure 3 (c) quantifies oscillation

Figure 3: **SWT decomposition of example token embeddings.** (a) $A_2$ capture smooth global trends, while (b) $D_2$, $D_1$ fluctuate with increasing oscillation at higher frequencies. (c) Distributional statistics for local oscillation strength across the sequence.

strength : for each sequence, we compute the first-order difference of coefficients over time and then evaluate the standard deviation within a $\omega$-width sliding window (practically, we used $\omega = 7$). A larger value means the signal fluctuates more strongly within that local region, reflecting higher oscillation strength. Together, the plots indicate that $A_2$ provides stable global structure, $D_2$ carries mid-scale details, and $D_1$ concentrates the most volatile fluctuations, which motivates the need to assess high-frequency ($D_1$) reliability before using it for adaptation.

## 3.2 HIGH-FREQUENCY SALIENCY

High-frequency coefficients $D_1$ capture local fine-grained variation that holds the details, but also include noise. To quantify their strength in a robust manner, we apply a soft shrinkage relative to a per-channel noise scale $\sigma_c$, a MAD-derived statistic computed along the token axis. This produces denoised details

$$\tilde{D}_1(t, c) \ = \ \text{sign}\big(D_1(t, c)\big) \ \text{ReLU}\big(|D_1(t, c)| - \sigma_c\big).$$

We then compute normalized token-wise energy and map it into $[0, 1]$ with a sigmoid,

$$E(t) \ = \ \sigma\Big(\frac{1}{D} \sum_{c=1}^{D} \frac{\tilde{D}_1(t, c)^2}{\sigma_c^2 + \varepsilon}\Big), \qquad \sigma(u) = \frac{1}{1 + e^{-u}}.$$

This yields a high-frequency saliency score such that tokens with strong, non-trivial energy across feature channels receive higher $E(t)$.

### 3.3 PARENT-GUIDED RELIABILITY

Saliency alone may respond to unstable spikes, which can add noisy perturbation to semantically meaningful high-frequency representations. To capture stability, we exploit the persistence of coefficients across scales. Using level-2 details $D_2$ as reference coefficients, we compute the inter-scale agreement as

$$r(t,c) = \frac{2D_1(t,c)D_2(t,c)}{D_1(t,c)^2 + D_2(t,c)^2 + \varepsilon\sigma_c^2}, \qquad g(t,c) = \tanh(\gamma r(t,c)),$$

with sharpness control hyperparameter $\gamma > 0$. The agreement score $g(t,c)$ takes values in $[-1,1]$ and reflects both the sign and relative magnitude of $D_1$ and $D_2$. The score becomes stronger when child ($D_1$) and parent ($D_2$) coefficients are aligned in both sign and magnitude and weaker when they are misaligned. However, values of $g$ near zero lie in the steep, unsaturated region of the $\tanh$ nonlinearity, where small changes in $r$ cause large fluctuations in $g$, making such scores unreliable.

To account for the instability of $g$ around zero, we further introduce a light-weight computable confidence term $\mathcal{C}(t,c)$ derived from the local sensitivity of $g$ w.r.t. $r$. Since

$$\frac{\partial g}{\partial r}(t,c) = \gamma \operatorname{sech}^2(\gamma r(t,c)),$$

large derivatives occur when $g$ is close to $0$, indicating unstable response to small changes in $r$. Conversely, when $g$ saturates near $\pm 1$, the derivative vanishes, indicating strong confidence in the agreement. We therefore define the confidence score as

$$\mathcal{C}(t,c) = 1 - \frac{1}{\gamma}\left(\frac{\partial g}{\partial r}(t,c)\right) = 1 - \operatorname{sech}^2(\gamma r(t,c)) = \tanh^2(\gamma r(t,c)),$$

which maps sensitivity into $[0,1]$. The final parent-guided reliability for each token $t$ is then given by

$$R(t) = \frac{1}{D}\sum_{c=1}^{D}\underbrace{\sigma(g(t,c))}_{\text{agreement}} \cdot \underbrace{\mathcal{C}(t,c)}_{\text{confidence}},$$

This reliability is high when child coefficients are consistent with their parent and the gate is confident (saturated), selectively emphasizing cross-scale supported features.

Figure 4 illustrates whether the saliency and reliability measures behave in accordance with their intended design. For this, we define per-channel token scores:

$$E(t,c) = \sigma\left(\frac{\tilde{D}_1(t,c)^2}{\sigma_c^2 + \varepsilon}\right), \qquad R(t,c) = \sigma(g(t,c)) \cdot \mathcal{C}(t,c).$$

For visualization purposes, $E(t,c)$ and $R(t,c)$ are further normalized to the range $[0,1]$. In panel (a), each scatter point corresponds to a coefficient pair $(D_1(t,c), D_2(t,c))$, colored by its saliency $E(t,c)$. Points with larger $D_1(t,c)$ map to higher saliency, confirming that $E(t,c)$ captures the intensity of fine-scale variations. In panel (b), reliability $R(t,c)$ is maximized when $D_1(t,c)$ and $D_2(t,c)$ are aligned in both sign and magnitude, and weakest off-diagonal when they disagree. Together, these plots show that $E(t,c)$ highlights token-level detail, while $R(t,c)$ refines it by emphasizing cross-scale consistent activations and suppressing unstable ones.

### 3.4 TOKEN-WISE SCORE AND LORA INTEGRATION

As a final token-wise gating score, we combine saliency and reliability through a geometric mean,

$$S(t) = \left(E(t)\right)^{1-\lambda}\left(R(t)\right)^{\lambda}, \qquad \lambda \in [0,1],$$

and map this token-level score to a scaling factor $s(t) = \sigma(S(t))$ that modulates the LoRA residual path. As our baseline LoRA structure, we adopt orthogonal LoRA (O-LoRA) (Wang et al., 2023a), which assigns each task a separate LoRA subspace while freezing those from previous tasks, and imposes cross-task orthogonality constraints to reduce subspace overlap across tasks. This provides a strong and widely used starting point for LoRA-based continual learning.

On top of this baseline, TUNE introduces token-dependent rescaling of the LoRA residual updates:

$$y_{\text{out}}(t) = Wx(t) + s(t)\,\Delta W_{\text{LoRA}}(x(t); \theta_k),$$

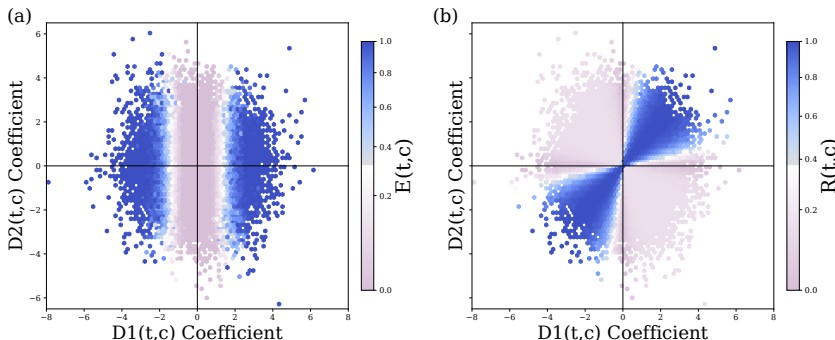

Figure 4: **Scatter plots of normalized saliency** $E(t,c)$ **and reliability** $R(t,c)$. Each point corresponds to a coefficient pair $(D_1(t,c), D_2(t,c))$, colored by its normalized score in $[0,1]$. (a) Larger $D_1(t,c)$ values generally correspond to higher $E(t,c)$. (b) $R(t,c)$ is strongest along the diagonal where $D_1(t,c)$ and $D_2(t,c)$ align in both sign and magnitude, and weak off-diagonal.

where $\theta_k$ denotes the learnable set of LoRA parameters for the current task $k$. This design provides two complementary benefits. During the forward pass, the scaling factor $s(t)$ adjusts the contribution of each token, effectively regulating its influence in the residual representations and thereby controlling token-wise attention to salient versus noisy signals. During the backward pass, treating $s(t)$ as constant with respect to $\theta_k$ yields

$$\frac{\partial \mathcal{L}}{\partial \theta_k} = \sum_{t=1}^{L} \underbrace{s(t)}_{\text{token-dependent LR}} \frac{\partial \mathcal{L}}{\partial y_{\text{out}}(t)} \frac{\partial \Delta W_{\text{LoRA}}(x(t); \theta_k)}{\partial \theta_k}.$$

Here $s(t)$ acts as a token-dependent learning rate (LR), scaling the gradient update contributed by each token according to its saliency and reliability. Tokens with high $s(t)$ exert stronger influence on parameter updates, promoting plasticity for novel yet reliable signals. In contrast, tokens with low $s(t)$ contribute little, helping to preserve stability against noisy or unstable activations. In this way, the scale factor regulates not only the forward significance of each token in residual adaptation but also its backward learning step size, effectively controlling how spectral cues shape the stability–plasticity trade-off during optimization.

Note that TUNE adds no additional learnable parameters beyond the continual expansion of LoRA modules already inherent to the O-LoRA framework. The scaling factors $s(t)$ are computed directly from spectral cues given by SWT decomposition of token embeddings without any parameterization. A detailed analysis of the resulting computational overhead is provided in the next section.

## 4 EXPERIMENTS

### 4.1 DATASETS

We evaluate TUNE under the widely used continual learning benchmarks for LLMs. The Standard CL benchmark consists of four text classification tasks (AG News, Amazon, Yelp, DBPedia, Yahoo) arranged in three different orders. For a greater challenge on longer sequence of continual tasks, we also adopt a Large-scale CL benchmark comprising fifteen different tasks: the five standard CL benchmark tasks, four GLUE tasks (MNLI, QQP, RTE, SST-2), five SuperGLUE tasks (WiC, CB, COPA, MultiRC, BoolQ), and IMDB reviews, also in three different orders. All task instructions follow the unified instruction-tuning schema from prior works (Qin & Joty, 2022; Wang et al., 2023a; Qiao & Mahdavi, 2024; Yang et al., 2024).

### 4.2 MODELS AND TRAINING

Experiments are conducted on two representative LLMs: the encoder–decoder T5-large (Raffel et al., 2023) and the decoder-only LLaMA-7B (Touvron et al., 2023). LoRA modules are inserted into the query and value projection matrices, while pretrained weights are frozen. For continual learning, we adopt O-LoRA (Wang et al., 2023a) as the baseline framework. TUNE is integrated into this baseline by introducing token-dependent rescaling of the LoRA residual updates. Unless otherwise noted, we

Table 1: Testing performance on two standard CL benchmarks with T5-large.[1] from Wang et al. (2023a), [2] from Qiao & Mahdavi (2024), [3] reproduced by us.

| Method | Standard CL Benchmark | | | | Large Number of Tasks | | | |
|---|---|---|---|---|---|---|---|---|
| | Order-1 | Order-2 | Order-3 | avg | Order-4 | Order-5 | Order-6 | avg |
| ProgPrompt[1] | 75.2 | 75.0 | 75.1 | 75.1 | 78.0 | 77.7 | 77.9 | 77.9 |
| PerTaskFT[1] | 70.0 | 70.0 | 70.0 | 70.0 | 78.1 | 78.1 | 78.1 | 78.1 |
| MTL[1] | 80.0 | 80.0 | 80.0 | 80.0 | 76.5 | 76.5 | 76.5 | 76.5 |
| SeqFT[1] | 18.9 | 24.9 | 41.7 | 28.5 | 7.4 | 7.4 | 7.5 | 7.4 |
| IncLoRA[1] | 66.0 | 64.9 | 68.3 | 66.4 | 63.3 | 58.5 | 61.7 | 61.2 |
| Replay[1] | 55.2 | 56.9 | 61.3 | 57.8 | 55.0 | 54.6 | 53.1 | 54.2 |
| EWC[1] | 48.7 | 47.7 | 54.5 | 50.3 | 45.3 | 44.5 | 45.6 | 45.1 |
| L2P[1] | 60.3 | 61.7 | 61.1 | 60.7 | 57.5 | 53.8 | 56.9 | 56.1 |
| LFPT5[1] | 67.6 | 72.6 | 77.9 | 72.7 | 70.4 | 68.2 | 69.1 | 69.2 |
| O-LoRA[1] | 75.4 | 75.7 | 76.3 | 75.8 | 72.3 | 64.8 | 71.6 | 69.6 |
| LB-CL[2] | 76.9 | 76.5 | 76.8 | 76.7 | 68.4 | 67.3 | 71.8 | 69.2 |
| N-LoRA[3] | 76.0 | 78.1 | 77.6 | 77.2 | 74.3 | 69.4 | 64.8 | 69.5 |
| **TUNE** | **78.5** | **78.4** | **78.0** | **78.3** | **75.4** | **73.0** | **75.9** | **74.8** |

follow the same experimental configurations of O-LoRA with 1-epoch training for every task except the learning rate and TUNE-specific hyperparameters. All experiments were conducted on NVIDIA A6000 GPUs, using the DeepSpeed framework for implementation. For evaluation, performance is measured by Average Accuracy (AA) after training the final task, i.e., $AA = \frac{1}{T}\sum_{i=1}^{T} a_{i,T}$.

## 4.3 BASELINES

We compare TUNE against a broad set of baselines commonly used in continual learning with LLMs. Independent training methods include **PerTaskFT**, which trains a separate model for each task, and **ProgPrompt** (Razdaibiedina et al., 2023), which learns independent prompts. As an optimistic upper bound, Multi-Task Learning (**MTL**) jointly trains on all tasks simultaneously. Non-continual fine-tuning methods include **SeqFT** (de Masson d'Autume et al., 2019), which updates all parameters on a sequence of tasks, and **IncLoRA**, an incremental learning of new LoRA parameters on a sequence of tasks without any regularization or replay. Regularization and replay-based approaches are also considered: **EWC** (Kirkpatrick et al., 2017) fine-tunes entire model with a regularization loss based on Fisher information and **Replay** maintains a buffer of prior samples for rehearsal. Prompt-based methods such as **L2P** (Wang et al., 2022) and **LFPT5** (Qin & Joty, 2022) dynamically select or generate prompts to adapt to new tasks. Finally, we compare against recent LoRA-based continual learning methods, including **O-LoRA** (Wang et al., 2023a), which enforces orthogonality between task-specific subspaces, **LB-CL** (Qiao & Mahdavi, 2024), which balances LoRA consolidation across tasks and **N-LoRA** (Yang et al., 2024), which reduces parameter collisions via sparsity.

Note that most of the baselines are taken from O-LoRA, as we adopt same experimental setting and codebase provided in their repository. For LB-CL, we report the results from the original paper due to the absence of publicly available code. For N-LoRA, we observe that they increase the number of training epochs per task from 1 (as used in O-LoRA and other baselines) to 10, which significantly inflates computational cost while yielding only marginal performance gains. To ensure consistency and fairness, we reproduce N-LoRA using the official codebase but adjust the training epoch to 1.

## 4.4 RESULTS

Tables 1 and 2 summarize the results on the standard and large-scale CL benchmarks with T5-large and LLaMA-7B. On the standard 4-task benchmark, TUNE achieves the highest average accuracy of 78.3, surpassing all other baselines. The improvement is more pronounced in the large-

Table 2: Testing performance on two standard CL benchmarks with LLaMA-7B. [1] reproduced by us.

| CL Benchmark | Order-1 | Order-2 | Order-3 | avg |
|---|---|---|---|---|
| O-LoRA[1] | 74.8 | 73.7 | 78.3 | 75.6 |
| N-LoRA[1] | 71.2 | 75.1 | 76.2 | 74.2 |
| **TUNE** | **77.8** | **78.1** | **79.8** | **78.6** |

scale 15-task benchmark, where TUNE reaches 74.8 as average accuracy, significantly higher than O-LoRA (69.6) and N-LoRA (69.5). These gains indicate that token-wise frequency-guided modulation provides better stability under long sequences of tasks where forgetting is most severe.

On LLaMA-7B in Table 2, where model scale makes forgetting even more challenging, TUNE again achieves the best performance. It improves the average accuracy on the standard benchmark to 78.6, compared to 75.6 for O-LoRA and 74.2 for N-LoRA. This demonstrates that the effectiveness of TUNE generalizes across architectures, showing consistent benefits in CL. Taken together, these results highlight that TUNE not only strengthens the stability–plasticity trade-off but also scales effectively to larger models and longer task sequences. By leveraging frequency cues to guide token-level adaptation, TUNE delivers SOTA performance among LoRA-based continual learning methods. We further analyze catastrophic forgetting using the Backward Transfer (BWT) metric, as detailed in Appendix H. The results show TUNE consistently reduces negative BWT compared to O-LoRA and N-LoRA, confirming its ability to better preserve prior knowledge across tasks.

**Computational Analysis.** TUNE introduces no additional trainable parameters beyond O-LoRA, as shown in Table 3, and incurs only negligible runtime overhead. The extra cost comes from the SWT decomposition and token-level scoring, both of which scale linearly with $BLD$. This cost is

Table 3: Comparison of training computation cost.

| Method | GPU Memory | Num of training params |
|--------|------------|------------------------|
| O-LoRA | 23.68 GB | $r(m+n)$ |
| LB-CL | 28.28 GB | $r(m+n)+r$ |
| N-LoRA | 23.69 GB | $r(m+n)$ |
| TUNE | 24.02 GB | $r(m+n)$ |

minor compared to the dominant Transformer operations such as quadratic attention $\mathcal{O}(BL^2D)$ and feed-forward layers $\mathcal{O}(BLD^2)$. A more detailed complexity analysis is provided in Appendix G.

## 4.5 ABLATION STUDIES

We conduct ablation studies to examine the impact of different design choices in TUNE. Detailed analyses on wavelet properties, scaling distributions, and additional variants are provided in Appendix D, E, and F. The main results are highlighted in Figure 5.

**Wavelet function** We adopt Symlets-8 (sym8) as the default wavelet for TUNE, as also reported in Table 1. Sym8 achieves the strongest performance due to its near-symmetry, which reduces phase distortion and improves cross-scale alignment, together with a moderate filter length that balances frequency resolution and temporal localization. Shorter filters such as Daubechies-4 (db4) are computationally cheaper but suffer

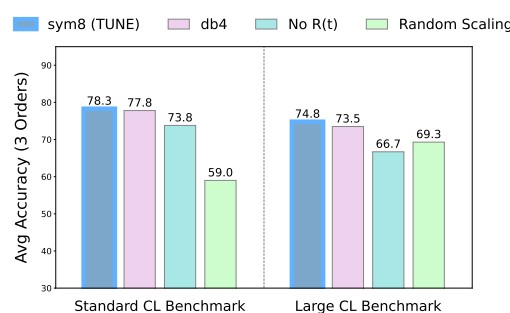

Figure 5: **Ablation results on TUNE.** Effect of wavelet function choice, removal of inter-scale reliability $R(t)$, and replacement with random scaling.

from stronger phase misalignment, resulting in a modest drop in accuracy. These findings show that wavelet choice matters, with sym8 offering the most consistent gains, while TUNE remains competitive across various filters and continues to outperform non-frequency baselines.

**No reliability regulation** We next ablate the inter-scale reliability term $R(t)$. Using saliency alone for gating tokens, i.e., $s(t) = E(t)$, causes a clear degradation in accuracy, demonstrating that reliability plays a critical role in filtering out unstable high-frequency activations.

**Random scaling** Finally, we replace TUNE's spectral scaling with random scaling factors drawn uniformly from the same empirical range of $s(t)$ (0.55–0.65) driven by TUNE with sym8 (see Appendix E). Despite matching the overall scale distribution, this variant performs substantially worse, indicating that the gains of TUNE cannot be attributed to random scaling of token activations. Instead, the improvements stem from the structured use of spectral cues—saliency and reliability—that adaptively modulate tokens in a task-sensitive manner.

## 5 CONCLUSION

We proposed TUNE, a frequency-guided token modulation method for continual learning in LLMs. By integrating wavelet-based saliency and reliability into token-wise LoRA updates, TUNE mitigates forgetting while enabling stable transfer. Across standard and large-scale benchmarks, it achieves consistent gains over strong baselines with no extra parameters and negligible overhead, establishing frequency-aware modulation as an effective principle for stable continual learning.

ETHICS STATEMENT.

This work does not involve human subjects, personal data, or sensitive attributes. The proposed method builds upon publicly available benchmark datasets (CL benchmark, GLUE, SuperGLUE, IMDB), all of which are widely used in the research community. Our contributions focus on algorithmic improvements for continual learning in large language models, with no foreseeable risks of ethical harm beyond those already present in standard LLM training.

REPRODUCIBILITY STATEMENT.

We have made all implementation details explicit, including hyperparameters, model configurations, datasets, and task sequences. Detailed training settings for both T5-large and LLaMA-7B are provided in the appendix (Tables 7, 8). All benchmark datasets are publicly accessible, and the exact task orders used for continual learning are documented in Appendix I.2. We will release the source code and scripts to fully reproduce the reported results upon publication.

THE USAGE OF LLMs.

In the preparation of this paper, we used large language models (LLMs) in a limited and supporting capacity. Specifically:

- **Writing aid and polishing:** LLMs were employed to improve the clarity, readability, and grammar of the manuscript. Their role was restricted to stylistic suggestions and refinement of phrasing, without altering the scientific content, claims, or conclusions.

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

## A    CODE AVAILABILITY

The source code for TUNE is available at `https://anonymous.4open.science/r/TUNE-C52F`

## B    FULL ALGORITHMIC DETAILS FOR TUNE

We present the full step-by-step procedure of TUNE. The algorithm outlines how token embeddings are decomposed by a two-level SWT, how saliency and reliability scores are computed, and how their fusion produces a token-wise scaling factor that modulates the LoRA residual updates. All input–output shapes are specified for clarity.

---

**Algorithm 1** TUNE for LoRA-residual scaling

---

1: **Input:** dataset of $N$ token sequences; stream of mini-batches $\{x^{(b)}\}_{b=1}^{N/B}$ where each $x^{(b)} \in \mathbb{R}^{B \times L \times D}$; two-level SWT filters $(h, g)$; hyperparameters $\gamma, \kappa, \lambda$; numeric $\varepsilon > 0$.

2: **for** each mini-batch $x^{(b)}$ **do**

3:     **2-Level SWT decomposition.**

4:     $(A_2, D_2, D_1) \leftarrow \text{SWT}(x^{(b)}; h, g, \text{level} = 2)$         $(A_2, D_2, D_1 \in \mathbb{R}^{B \times L \times D})$

5:     **High-frequency saliency $E(t)$.**

6:     $\tilde{D}_1 \leftarrow \text{sign}(D_1) \cdot \text{ReLU}(|D_1| - \sigma_c)$

7:     where $\sigma_c \in \mathbb{R}^{B \times D}$ is a per-channel noise scale (MAD statistic along tokens).

8:     $E(t) \leftarrow \sigma\left(\frac{1}{D} \sum_{c=1}^{D} \frac{\tilde{D}_1^2}{\sigma_c^2 + \varepsilon}\right)$         $(E \in [0,1]^{B \times L \times 1})$

9:     **Parent-guided reliability $R(t)$.**

10:     $r(t, c) \leftarrow \frac{2 D_1 \odot D_2}{D_1^2 + D_2^2 + \varepsilon \sigma_c^2}$         $(r \in \mathbb{R}^{B \times L \times D})$

11:     $g(t, c) \leftarrow \tanh(\gamma r); \quad \mathcal{C}(t, c) \leftarrow \tanh^2(\gamma r)$     $(g \in (-1,1)^{B \times L \times D}, \mathcal{C} \in (0,1)^{B \times L \times D})$

12:     $R(t) \leftarrow \frac{1}{D} \sum_{c=1}^{D} \sigma(\kappa g(t, c)) \odot \mathcal{C}(t, c)$     $(R \in [0,1]^{B \times L \times 1})$

13:     **Token score and scaling.**

14:     $S(t) \leftarrow E(t)^{1-\lambda} \odot R(t)^{\lambda}$

15:     $s(t) \leftarrow \sigma(S)$         $(s \in (0,1)^{B \times L \times 1})$

16:     **LoRA integration.**

17:     $y_{\text{out}}(t) \leftarrow W x^{(b)}(t) + s(t) \Delta_{\text{LoRA}}(x^{(b)}(t); \theta)$     $(y_{\text{out}} \in \mathbb{R}^{B \times L \times d_{\text{out}}})$

18: **end for**

---

## C    FREQUENCY-BAND ANALYSIS OF TOKEN REPRESENTATIONS

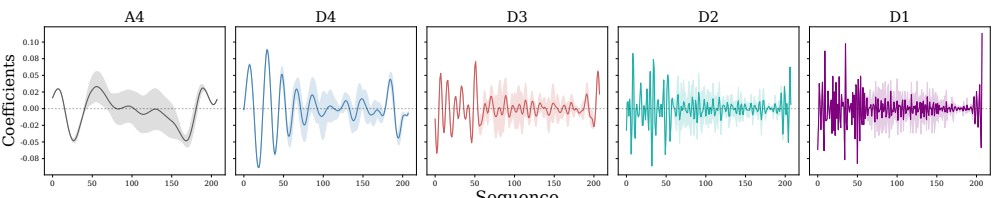

Figure 6: **Example level 4 SWT traces on DBPedia with sym8 filter.** The embedding sequence is decomposed into approximation ($A_4$) and detail components ($D_4$–$D_1$). Each subplot shows the mean coefficient trajectory across tokens with 95% confidence intervals, illustrating how different frequency bands emphasize smooth trends ($A_4$) versus progressively higher-frequency fluctuations ($D_4$–$D_1$).

To better understand how token representations are distributed across frequency bands, we decompose the embedding sequence using a four-level stationary wavelet transform (SWT). This yields one approximation component ($A_4$) and four detail components ($D_4$–$D_1$), which progressively capture lower- to higher-frequency variations. For each token position, the coefficients are averaged across embedding dimensions and aggregated over multiple samples to produce mean traces with confidence intervals. Figure 6 shows an SWT-decomposed example trace across scales from the DBPedia dataset using the Symlet-8 (sym8) wavelet filter, illustrating how different frequency bands emphasize distinct patterns of variation along the sequence.

## C.1 High-Frequency ($D_1$) Trace Adjusted with Reliability ($R(t)$)

To highlight the effect of $R(t)$, the reliability term, we plot the z–scored amplitude of the raw high-frequency coefficients $D_1(t)$ and their reliability–adjusted trace $D_1^{\mathrm{adj}}(t) = R(t) \cdot D_1(t)$. Although $R(t)$ is not directly used to alter high-frequency components of token embeddings in TUNE, this visualization illustrates how it effectively suppresses spiky, unstable variations in $D_1$.

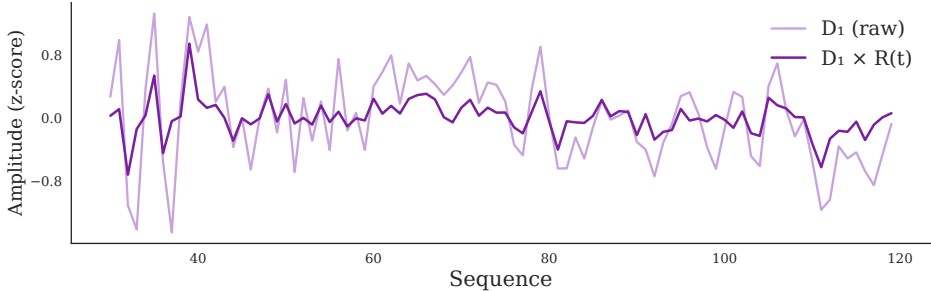

Figure 7: **High-frequency coefficients $D_1(t)$ with reliability adjustment.** Applying $R(t)$ to $D_1(t)$ suppresses erratic spikes in the raw $D_1(t)$ trace, yielding a smoother high-frequency profile.

## C.2 Task Discriminability of Frequency Bands

To examine how different frequency bands contribute to task separation, we probe the task discriminability of low- ($A_{10}$), mid- ($D_8$), and high-frequency ($D_1$) SWT coefficients on four tasks from the standard CL benchmark. For each band, we train a logistic regression classifier with 5-fold cross-validation to obtain a confusion matrix, and compute inter-task distances using Maximum Mean Discrepancy (MMD). This analysis reveals how reliably each band preserves task-specific information versus collapsing into overlapping noise.

## D Choice of Wavelet Function

Selecting an appropriate wavelet is crucial for effectively decomposing token embeddings in our setup. In practice, the choice is guided by several factors: (i) orthogonality, which ensures non-redundant coefficient representations and stable reconstruction; (ii) symmetry, which mitigates phase distortions and preserves temporal alignment of token features across scales; and (iii) filter length, which balances frequency resolution against temporal localization. Longer filters (e.g., Coiflets) capture finer spectral structure at the cost of reduced localization, while shorter filters (e.g., Daubechies-4) emphasize compact support but often introduce greater asymmetry.

Figure 9 visualizes the filters used in our experiments. **Symlets-8 (sym8)** are nearly symmetric with a moderate filter length ($K = 16$), ensuring that low-pass and high-pass filters remain phase-aligned. This property reduces distortions in cross-scale comparisons and makes reliability estimation more stable, a particularly desirable feature for language data where both local token ordering and global semantic consistency must be preserved. **Coiflets-5 (coif5)** have longer filters ($K = 30$) with vanishing moments for both scaling and wavelet functions, providing stronger frequency selectivity but weaker temporal localization. **Daubechies-8 (db8)** extend the Daubechies family to a longer filter length ($K = 16$), gaining smoothness at the cost of greater asymmetry. Finally, **Daubechies-4 (db4)** are short filters ($K = 8$) that are computationally efficient but exhibit stronger phase distortion

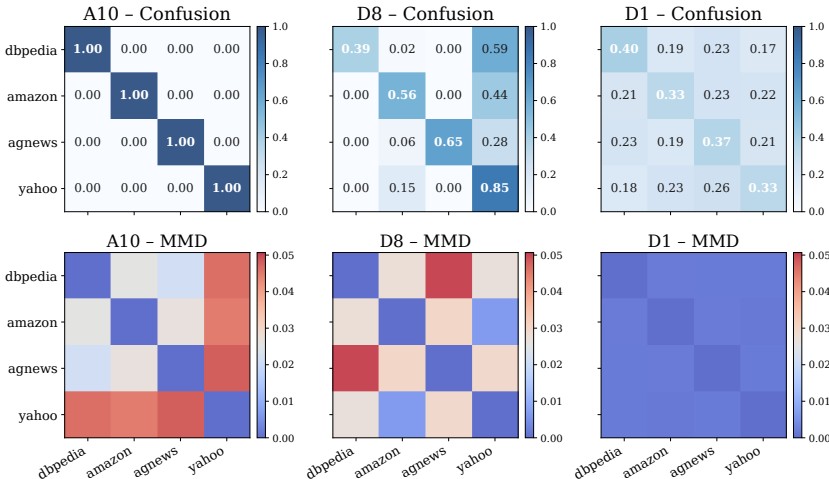

Figure 8: **Task discriminability across frequency bands.** Left: confusion matrices from a linear probe. Right: pairwise Maximum Mean Discrepancy (MMD) between tasks. $A_{10}$ exhibits nearly perfect separation, confirming that low-frequency components encode stable, task-specific semantics. $D_8$ shows partial separation, reflecting mid-frequency structure. In contrast, $D_1$ collapses into overlapping clusters with weak inter-task separation, highlighting its noisy and unstable nature.

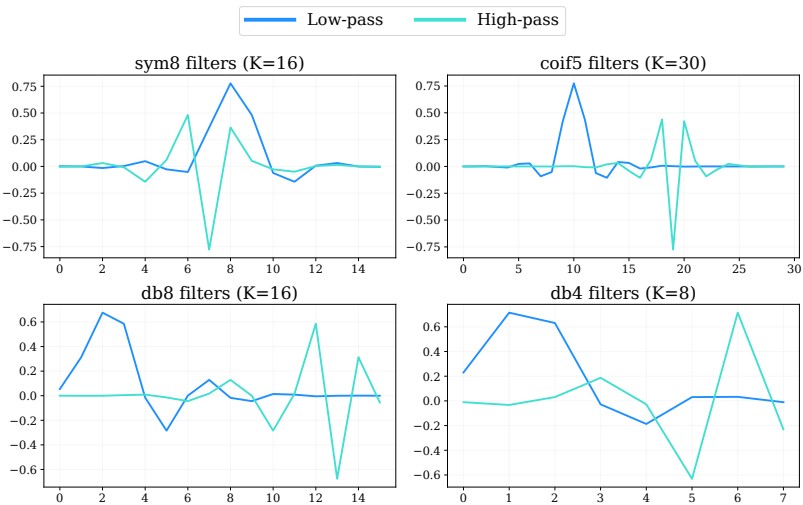

Figure 9: **Wavelet filters used in our experiments.** Low-pass (blue) and high-pass (green) decomposition filters for Symlets-8, Coiflets-5, Daubechies-8, and Daubechies-4. Sym8 is used as the default in the main results, while the others are considered in ablation studies.

compared to their longer counterparts. All wavelets considered (Symlets, Daubechies, Coiflets) are orthogonal families, ensuring energy-preserving, non-redundant filters.

We adopt sym8 as the default wavelet throughout the main experiments, as it provides the most effective balance between symmetry, frequency resolution, and temporal localization. Among these factors, symmetry is especially critical in our setup, since it minimizes phase distortions between low-and high-pass filters, ensuring stable cross-scale comparisons that underpin reliability estimation. This property is particularly important for language data, where local token structure and global semantic stability must be preserved simultaneously. Ablation studies with coif5, db8, and db4 (see Section F) further suggest that while performance can vary with filter choice, the overall method remains reasonably robust across different wavelet families.

# E    DISTRIBUTIONS OF $E(t)$, $R(t)$, AND $s(t)$

We visualize the pooled token–level distributions of the three measures used in our SWT analysis—energy/saliency $E(t)$, reliability $R(t)$ (the agreement score), and the combined scale $s(t)$—for the DBPedia task in standard CL benchmark. Values are aggregated across all batches and samples; each row corresponds to one wavelet family (sym8, coif5, db4), and each column to a measure. Across filters, $E(t)$ is right–skewed with most mass near 1.0 and a light mode around 0.6 (bimodal shape), indicating many tokens receive high saliency while a smaller group is moderate. This reflects the spiky and oscillatory behavior of high-frequency band. $R(t)$ concentrates in a mid range (around 0.13–0.56), presenting moderate but not extreme agreement. The scale $s(t)$ is narrow and stable (around 0.56–0.66) with minor shifts across filters, suggesting the combined scaling behaves consistently regardless of the wavelet choice. Vertical dashed lines mark the min/max of each pooled distribution.

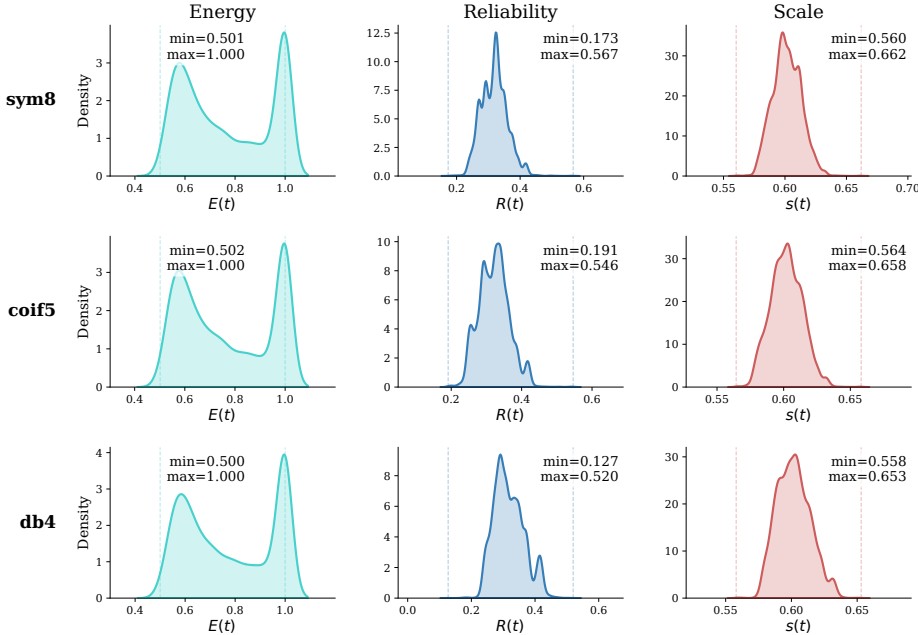

Figure 10: Density estimates of pooled token–level measures on DBPedia. Rows: wavelet filters (sym8, coif5, db4). Columns: saliency $E(t)$, reliability $R(t)$, and combined token scale $s(t)$. Vertical dashed lines denote the observed min/max for each distribution; y–axis is density.

# F    ABLATION STUDIES

Table 4: Testing performance on two standard CL benchmarks with T5-large.[1]:

| Method | Standard CL Benchmark | | | | Large Number of Tasks | | | |
|---|---|---|---|---|---|---|---|---|
| | Order-1 | Order-2 | Order-3 | avg | Order-4 | Order-5 | Order-6 | avg |
| TUNE + sym8 (Default) | **78.5** | **78.4** | **78.0** | **78.3** | **75.4** | **73.0** | **75.9** | **74.8** |
| TUNE + sym4 | 78.9 | 78.8 | 76.8 | 78.2 | 73.9 | 71.8 | 72.5 | 72.7 |
| TUNE + db8 | 79.0 | 79.1 | 78.0 | 78.7 | 73.0 | 71.3 | 73.1 | 72.5 |
| TUNE + db4 | 78.4 | 78.5 | 76.4 | 77.8 | 73.6 | 71.9 | 74.9 | 73.5 |
| TUNE + coif5 | 78.5 | 78.3 | 77.2 | 78.0 | 71.5 | 71.1 | 75.0 | 72.5 |
| TUNE w/o R(t) | 72.9 | 77.2 | 71.3 | 73.8 | 68.8 | 65.2 | 66.0 | 66.7 |
| Random Scaling | 51.5 | 49.8 | 75.8 | 59.0 | 72.2 | 65.2 | 72.0 | 69.3 |

We conduct ablation studies to examine the influence of various components of TUNE such as wavelet choice and the role of inter-scale reliability regulation. Results are summarized in Table 4.

For wavelet selection, we test four widely used families: Symlets-8 (sym8), Symlets-4 (sym4), Daubechies-8 (db8), Daubechies-4 (db4), and Coiflets-5 (coif5). Among them, sym8 achieves the highest overall performance, confirming that its balanced design—moderate filter length with near-symmetry—offers a particularly suitable trade-off for decomposing language embeddings. Nevertheless, other filters also yield strong results, showing that while the exact choice of wavelet impacts final accuracy, the frequency-guided token scaling mechanism remains consistently effective across families and outperforms non-frequency baselines.

We also ablate the reliability regulation term $R(t)$. Removing $R(t)$ and using saliency $E(t)$ alone for scaling produces a clear drop in accuracy, indicating that inter-scale reliability is crucial for suppressing unstable high-frequency activations. Finally, we replace TUNE-driven scaling with random scaling values drawn uniformly from $[0.55, 0.65]$, matching the empirical range of scales driven by TUNE (Figure 10). This variant performs far worse, highlighting that the gains of TUNE stem from principled spectral cues rather than arbitrary rescaling.

Together, these ablations confirm that frequency-guided scaling is a robust and indispensable mechanism, with sym8 serving as the most effective default wavelet for language continual learning.

## G    COMPUTATION ANALYSIS

We provide a detailed analysis of the additional computational cost introduced by TUNE. Let $B$ denote the batch size, $L$ the sequence length, $D$ the hidden size, $k$ the wavelet filter length, and $\ell$ the number of SWT decomposition levels.

**SWT Decomposition.**    At each level, the stationary wavelet transform (SWT) applies both a low-pass and a high-pass finite impulse response (FIR) filter. Thus, $\ell$ levels require $2\ell$ FIR convolutions in total. Each convolution costs $\mathcal{O}(BkLD)$, giving

$$T_{\text{SWT}} = \mathcal{O}(2\ell BkLD).$$

Since $k$ is small (e.g., $k \in [4, 16]$), this cost is linear in $BLD$ and negligible compared to quadratic attention or large matrix multiplications.

**Token Scoring.**    The computation of token saliency $E(t)$ and reliability $R(t)$ consists primarily of elementwise operations. For saliency, squaring the high-frequency coefficients, normalizing by per-channel variance, and applying a sigmoid costs $\mathcal{O}(BLD)$. Reliability involves computing the agreement between child–parent coefficients $(D_1, D_2)$, a sigmoid, and a lightweight confidence term $\mathcal{C}(t, c)$, also costing $\mathcal{O}(BLD)$. Finally, combining saliency and reliability into the token scaling factor $s(t)$ requires only a few elementwise operations, adding $\mathcal{O}(BL)$.

Summing the above contributions, the added complexity of TUNE is $T_{\text{TUNE}} = \mathcal{O}(2\ell BkLD) + \mathcal{O}(BLD)$, both terms scaling linearly with $BLD$ and involving only small constants. In contrast, the dominant Transformer costs are

$$T_{\text{attn}} = \mathcal{O}(BL^2D), \qquad T_{\text{FFN}} = \mathcal{O}(BLD^2),$$

which grow quadratically with $L$ or $D$. In contrast, the additional terms introduced by TUNE only scale linearly with $BLD$ and involve small constants tied to the filter length $k$ and decomposition depth $\ell$. Since $k \ll D$ and $\ell$ is typically small in practice, it follows that

$$T_{\text{TUNE}} \ll T_{\text{attn}}, \ T_{\text{FFN}},$$

indicating that the extra cost of TUNE is asymptotically insignificant compared to the dominant Transformer operations. This conclusion is further supported by the empirical memory usage comparison in Table 3, which confirms that TUNE adds negligible overhead while preserving the efficiency of the O-LoRA framework.

# H  CATASTROPHIC FORGETTING MEASURED BY BACKWARD TRANSFER (BWT)

Backward Transfer (BWT) quantifies catastrophic forgetting by measuring how much the accuracy on earlier tasks degrades after training on subsequent tasks. Formally, it is defined as

$$BWT = \frac{1}{T-1} \sum_{i=1}^{T-1} (a_{i,T} - a_{i,i}),$$

where $a_{i,j}$ denotes the test accuracy on the $i$-th task after training up to the $j$-th task. A negative BWT value indicates forgetting, with larger magnitudes corresponding to more severe degradation.

Table 5: Backward Transfer (BWT) across task orders on T5-Large.

| Method | Orders 1–3 | | | | Orders 4–6 | | | |
|--------|------|--------|-------|--------|-------|--------|-------|--------|
| | 1 | 2 | 3 | Avg | 4 | 5 | 6 | Avg |
| O-LoRA | -2.15 | -12.55 | -7.80 | -7.50 | -9.98 | -8.02 | -6.67 | -8.22 |
| N-LoRA | -2.68 | 1.00 | -0.17 | **-0.62** | -4.89 | -15.09 | -9.56 | -9.85 |
| **TUNE** | -0.08 | -1.60 | -1.04 | -0.91 | -5.24 | -8.74 | -3.89 | **-5.95** |

Table 6: Backward Transfer (BWT) for task orders 1–3 on LLaMA-7B.

| Method | Orders 1–3 | | | |
|--------|-------|-------|-------|--------|
| | 1 | 2 | 3 | Avg |
| O-LoRA | -14.65 | -8.70 | -3.77 | -9.04 |
| N-LoRA | -8.69 | -0.57 | -0.34 | -3.20 |
| **TUNE** | -1.29 | -3.56 | -1.61 | **-2.15** |

Before delving into the details, we note that all reported BWT values are computed from the same models whose performance is summarized in Table 1.

Table 5 reports BWT across different task orders on T5-Large. Compared to O-LoRA and N-LoRA, TUNE consistently achieves less negative BWT, indicating reduced forgetting. On Orders 1–3, TUNE reaches an average of $-0.91$, close to stable retention, while O-LoRA suffers a much larger drop of $-7.50$. Although N-LoRA achieves a slightly lower average ($-0.62$) on Orders 1–3, it deteriorates significantly in later tasks (Orders 4–6), where its BWT falls to $-9.85$. In contrast, TUNE maintains a substantially smaller degradation ($-5.95$), confirming its robustness as tasks accumulate. These results highlight that frequency-aware scaling not only preserves early tasks but also mitigates long-term forgetting across extended sequences.

Table 6 presents the same analysis on LLaMA-7B for Orders 1–3. Here the trend is even clearer: O-LoRA exhibits severe forgetting ($-9.04$ on average), while N-LoRA performs better ($-3.20$) but still suffers from unstable retention across tasks. TUNE consistently achieves the best balance with an average of $-2.15$, showing substantially reduced catastrophic forgetting compared to both baselines. Together, these results across two architectures confirm that TUNE's frequency-guided modulation provides a principled mechanism for alleviating forgetting, improving both short-term stability and long-term robustness.

# I  EXPERIMENTAL SETUP

## I.1  HYPERPARAMETERS

For all experiments, we follow the same training configurations as O-LoRA (Wang et al., 2023a), ensuring a fair comparison. The only differences are the learning rate, which we adjust for stability, and the TUNE-specific hyperparameters $(\gamma, \kappa, \lambda)$. Importantly, these hyperparameters are kept

fixed across all tasks and orders without per-task tuning. This consistency indicates that TUNE is robust to hyperparameter selection: the chosen values provide a balanced integration of token saliency and reliability into the final scaling factor, without requiring task-specific adjustment. Such stability is advantageous in continual learning, where excessive hyperparameter tuning across tasks would undermine practicality. Tables 7 and 8 summarize the detailed settings used for T5-large and LLaMA-7B experiments, respectively.

Table 7: Detailed training hyper-parameters and configuration for continual learning on T5-Large.

| Config | Order-1 | Order-2 | Order-3 | Order-4 | Order-5 | Order-6 |
|---|---|---|---|---|---|---|
| *TUNE Configs* | | | | | | |
| $\gamma$ | 2.0 | 2.0 | 2.0 | 2.0 | 2.0 | 2.0 |
| $\kappa$ | 2.0 | 2.0 | 2.0 | 2.0 | 2.0 | 2.0 |
| $\lambda$ | 0.7 | 0.7 | 0.7 | 0.7 | 0.7 | 0.7 |
| *O-LoRA Configs* | | | | | | |
| lambda 1 | 0.5 | 0.5 | 0.5 | 0.5 | 0.5 | 0.5 |
| lambda 2 | 0.0 | 0.0 | 0.0 | 0.0 | 0.0 | 0.0 |
| lora rank per task | 8 | 8 | 8 | 8 | 8 | 8 |
| lora dropout | 0.1 | 0.1 | 0.1 | 0.1 | 0.1 | 0.1 |
| lora target modules | Query, Value projection | | | | | |
| *General Configs* | | | | | | |
| Epoch | 1 | 1 | 1 | 1 | 1 | 1 |
| Learning rate | 1e-3 | 5e-4 | 4e-4 | 1e-3 | 5e-4 | 4e-4 |
| Gradient Accumulation Step | 1 | 1 | 1 | 1 | 1 | 1 |
| Train Batch size / GPU | 8 | 8 | 8 | 8 | 8 | 8 |
| Eval Batch size / GPU | 128 | 128 | 128 | 128 | 128 | 128 |
| GPU (A6000 48G) | 1 | 1 | 1 | 4 | 4 | 4 |

Table 8: Detailed training hyper-parameters and configuration for continual learning on LLaMA-7B.

| Config | Order-1 | Order-2 | Order-3 |
|---|---|---|---|
| *TUNE Configs* | | | |
| $\gamma$ | 2.0 | 2.0 | 2.0 |
| $\kappa$ | 2.0 | 2.0 | 2.0 |
| $\lambda$ | 0.7 | 0.7 | 0.7 |
| *O-LoRA Configs* | | | |
| lambda 1 | 0.5 | 0.5 | 0.5 |
| lambda 2 | 0.0 | 0.0 | 0.0 |
| lora rank per task | 8 | 8 | 8 |
| lora dropout | 0.1 | 0.1 | 0.1 |
| lora target modules | Query, Value projection | | |
| *General Configs* | | | |
| Epoch | 1 | 1 | 1 |
| Learning rate | 5e-4 | 5e-4 | 5e-4 |
| Gradient Accumulation Step | 8 | 8 | 8 |
| Train Batch size / GPU | 1 | 1 | 1 |
| Eval Batch size / GPU | 16 | 16 | 16 |
| GPU (A6000 48G) | 4 | 4 | 4 |

## I.2 DATASETS AND CL TASK SEQUENCES

We evaluate on 15 datasets spanning diverse domains and tasks, including sentiment classification, topic classification, natural language inference, and question answering. These datasets are drawn from established benchmarks: the standard CL benchmark with five tasks (Yelp, Amazon, DBpedia, Yahoo, AG News), GLUE (MNLI, QQP, RTE, SST-2), SuperGLUE (WiC, CB, COPA, MultiRC, BoolQ), and IMDB reviews. Table 9 summarizes their categories, tasks, and domains.

To assess both short- and long-horizon continual learning, we follow prior work (Wang et al., 2023a; Qiao & Mahdavi, 2024; Yang et al., 2024) and construct two settings. The Standard CL benchmark includes four tasks (AG News, Amazon, Yahoo, DBPedia) arranged in three different orders. The Large-scale CL benchmark expands to fifteen tasks by adding four GLUE tasks, five SuperGLUE tasks, and IMDB, also evaluated under three task sequences. Table 10 details the task orders used in our experiments.

Table 9: Datasets and Their Tasks, Categories, Domains, and Metrics

| No. | Dataset name | Category | Task | Domain | Metric |
|-----|-------------|----------|------|--------|--------|
| 1 | Yelp | CL Benchmark | sentiment analysis | Yelp reviews | accuracy |
| 2 | Amazon | CL Benchmark | sentiment analysis | Amazon reviews | accuracy |
| 3 | DBpedia | CL Benchmark | topic classification | Wikipedia | accuracy |
| 4 | Yahoo | CL Benchmark | topic classification | Yahoo Q&A | accuracy |
| 5 | AG News | CL Benchmark | topic classification | news | accuracy |
| 6 | MNLI | GLUE | NLI | various | accuracy |
| 7 | QQP | GLUE | paragraph detection | Quora | accuracy |
| 8 | RTE | GLUE | NLI | news, Wikipedia | accuracy |
| 9 | SST-2 | GLUE | sentiment analysis | movie reviews | accuracy |
| 10 | WiC | SuperGLUE | word sense disambig. | lexical databases | accuracy |
| 11 | CB | SuperGLUE | NLI | various | accuracy |
| 12 | COPA | SuperGLUE | QA | blogs, encyclopedia | accuracy |
| 13 | BoolQA | SuperGLUE | boolean QA | Wikipedia | accuracy |
| 14 | MultiRC | SuperGLUE | QA | various | accuracy |
| 15 | IMDB | SuperGLUE | sentiment analysis | movie reviews | accuracy |

Table 10: Task Sequences for Continual Learning

| Order | Model | Task Sequence |
|-------|-------|---------------|
| 1 | T5, LLaMA | dbpedia → amazon → yahoo → ag |
| 2 | T5, LLaMA | dbpedia → amazon → ag → yahoo |
| 3 | T5, LLaMA | yahoo → amazon → ag → dbpedia |
| 4 | T5 | mnli → cb → wic → copa → qqp → boolqa → rte → imdb → yelp → amazon → sst-2 → dbpedia → ag → multirc → yahoo |
| 5 | T5 | multirc → boolqa → wic → mnli → cb → copa → qqp → rte → imdb → sst-2 → dbpedia → ag → yelp → amazon → yahoo |
| 6 | T5 | yelp → amazon → mnli → cb → copa → qqp → rte → imdb → sst-2 → dbpedia → ag → yahoo → multirc → boolqa → wic |

