# OpenReview forum: "TUNE: Frequency-Guided Token Gating for Robust Continual Learning in LLMs"
_ICLR.cc/2026/Conference — ICLR 2026 Conference Withdrawn Submission_

### Official Review · Reviewer_5mKV · 2025-10-29

**Soundness:** 1
**Presentation:** 3
**Contribution:** 2
**Rating:** 2
**Confidence:** 3

**Summary:**

This paper proposes a new training method for LLMs, TUNE. TUNE decomposes gradients into per-token components and then applies a wavelet transform to extract contributions varying with different frequencies. After extracting fast and slow-varying components, they are combined with a weighted geometric mean. On T5-Large and Llama-7B, improvements are shown on sequences of 3-6 tasks drawn from GLUE/SuperGLUE/IMDB/etc. over baselines such as O-LoRA. Finally, ablation studies show the importance of the wavelet transform, the reliability term, and the scaling factors used by TUNE.

**Strengths:**

1. The idea of decomposing gradients into fast and slow-varying components is appealing. I think this direction can also be used for analysis beyond the training of models.
2. The math in this paper is presented well, and the figures are nice. I did not have much trouble following the details.
3. I find the ablation studies convincing and informative.

**Weaknesses:**

I find the following aspects weak in the current draft.
1. The expressions for reliability, saliency, and the final score are very complex. I would understand their use for analysis, but considering that a general training method for continual learning is being proposed here, these expressions are too hand-engineered.
2. The evaluation section is currently unconvincing. Here are my reasons as to why:

    a. The language models being used—T5-large and Llama (1)—are very old. Why not use modern language models such as Llama-3.1?

    b. Even though these models are old, GLUE tasks are very simple for them. I would expect Llama to perform very impressively on GLUE even with minimal fine-tuning. Therefore, evaluating on GLUE is not very informative.

    c. GLUE mainly consists of subsets where the model needs to “perform a task”, i.e., classify sentiment, translate examples, and so on. It is better to evaluate on tasks requiring memorization—and of obscure or synthetic facts—so that the abilities of the pretrained model do not confuse the results. For example, you could consider using a synthetic biography dataset like in [1].
3. As per Table 1, it appears that ProgPrompt and MTL outperform TUNE. The paper says that MTL is an upper bound since it trains on all examples simultaneously—which makes sense as a general argument, but is less convincing here because of the nature of the GLUE tasks described above.
4. (This point did not affect my score) The introduction section’s writing reads like related work. While I think the related works are relevant and nicely described, I believe the introduction should offer a more descriptive narrative (rather than place the paper among other related work).

[1] “Physics of Language Models: Part 3.1, Knowledge Storage and Extraction,” Allen-Zhu and Li, International Conference on Machine Learning (ICML) 2024

**Questions:**

I have the following questions for the authors to clarify my position and resolve my concerns with the current version.
1. What were the reasons for using T5/Llama-1 in the experiments? Do you have any results with newer models?
2. Are there any results or evaluations on knowledge-intensive tasks (could be synthetic)?

---

### Official Review · Reviewer_V2B5 · 2025-11-01

**Soundness:** 2
**Presentation:** 3
**Contribution:** 2
**Rating:** 2
**Confidence:** 3

**Summary:**

The paper investigates continual learning (CL) in large language models (LLMs) and introduces TUNE (Token Update via Noise-robust Frequency Encoding). The authors apply a stationary wavelet transform (SWT) to decompose embeddings into multi-resolution subbands. They derive token-wise saliency and reliability scores to modulate LoRA residual updates, amplifying stable signals and suppressing noisy ones. Without adding trainable parameters, TUNE reportedly improves performance across standard CL benchmarks (Amazon, Yelp, DBPedia, Yahoo, etc). The work positions frequency-domain regularization as a new perspective for stabilizing continual adaptation in LLMs.

**Strengths:**

1. The paper introduces a frequency-domain interpretation of catastrophic forgetting in LLMs, which is conceptually original and offers an alternative to conventional rehearsal or regularization approaches.
2. The writing is clear and easy to follow.
3. Experimental results demonstrates the effectiveness of this method.

**Weaknesses:**

1. The proposed use of stationary wavelet transform (SWT) on token embeddings does not rigorously satisfy the mathematical properties required for a valid wavelet analysis. While SWT enforces shift invariance and zero-mean filtering, it lacks orthogonality, completeness, and exact reconstructibility in the high-dimensional embedding space. Consequently, the resulting decomposition is not a true wavelet transform but a heuristic multi-scale filtering operation that cannot guarantee information preservation or invertibility. For example, how are the basis functions defined here, and what is their physical meaning?
2. The paper attributes catastrophic forgetting primarily to overfitting on spurious or noisy high-frequency patterns (lines 144–123). However, this claim appears overly strong. Forgetting in continual learning cannot be fully explained by overfitting alone. One plausible explanation is the intrinsic non-convexity of neural network optimization landscapes. As the model adapts to each new task, the optimizer converges toward a different local minimum, and the resulting shift in the basin of attraction naturally alters previously learned representations. ' Overfitting on spurious pattern' is not able cover most of scenarios.
3. Similiarly, the datasets employed (Amazon, Yelp, DBPedia, Yahoo) are relatively generic and fail to capture the domain-specific complexity where catastrophic forgetting is most pronounced, such as in specialized areas like medicine or law. Incorporating domain-specific continual-learning benchmark, such as EHRShot, MIMIC-III or other domain-incremental datasets would substantially strengthen the empirical evaluation and demonstrate the method’s robustness in realistic, knowledge-intensive settings.

**Questions:**

see above

---

### Official Review · Reviewer_jZ5k · 2025-11-01

**Soundness:** 2
**Presentation:** 3
**Contribution:** 2
**Rating:** 4
**Confidence:** 4

**Summary:**

This paper proposes TUNE: frequency-guided token Gating for robust continual learning in LLMs. The main ideas of this paper are high-frequency saliency, parent-guided reliability, and token-wise score and LoRA integration. While discussing an interesting topic and delivering a good presentation, the paper has several issues that need to be addressed. Please see the weaknesses.

**Strengths:**

(1). The paper discusses a challenging problem.

(2). The paper has a good presentation and is easy to follow.

(3). The paper presents a rigorous numerical analysis.

**Weaknesses:**

(1). Methodology: frequency-guided LoRA for continual LLM is arguably not a novel idea; previous work, i.e., MultLFG, presented a similar idea.

(2). Performance: The proposed method achieves only a small margin for some cases in both of the benchmarks, i.e., Order-2, Order-3, Order-4 (Table 1). It is questionable that the proposed method significantly outperforms the existing methods.

(3). Forgetting: I do not see the measurements and analysis of models' forgetting as the answer to the catastrophic forgetting problem.

(4). Theoretical Analysis: I do not see a solid theoretical foundation or theoretical analysis (proof) of the proposed method.

(5). Missing comparison to the up-to-date methods, e.g., Multi-Stage-Learning (NAACL 2025), I-LoRA (ICASSP 2025)




References:

[1]. MultLFG: Training-free Multi-LoRA composition using Frequency-domain Guidance

[2]. Multi-Stage-Learning: Multi-Stage LLM Fine-Tuning with a Continual Learning Setting

[3]. I-LoRA: Analyzing and Reducing Catastrophic Forgetting in Parameter-Efficient Tuning

**Questions:**

Please address the weaknesses.

---

### Official Review · Reviewer_c336 · 2025-11-01

**Soundness:** 3
**Presentation:** 2
**Contribution:** 3
**Rating:** 4
**Confidence:** 4

**Summary:**

This paper addresses catastrophic forgetting in continual learning of LLMs by introducing TUNE, a frequency-guided token modulation mechanism integrated with LoRA. TUNE uses a Stationary Wavelet Transform (SWT) to decompose token embeddings into multi-resolution subbands.

**Strengths:**

1. This is the first work to explicitly link high-frequency drift to catastrophic forgetting in LLMs’ CL, filling a gap in prior work that ignored spectral properties. SWT-based decomposition effectively separates stable (low-frequency) and noisy (high-frequency) signals, providing a principled basis for mitigating forgetting.

2. TUNE adds no extra trainable parameters beyond LoRA, matching O-LoRA/N-LoRA in parameter count while introducing negligible overhead (linear scaling with batch/sequence size.

3. By combining saliency and reliability for token gating, TUNE reduces negative Backward Transfer (BWT).

**Weaknesses:**

1. Only evaluates classification tasks (sentiment, topic, NLI, QA). It does not test complex tasks like open-ended generation (summarization, dialogue) or reasoning (math, logic), where token-level frequency cues may behave differently.

2. Key hyperparameters are fixed across tasks but lack empirical or theoretical justification.

3. Only tested on T5 and LLaMA2. It is unknown if TUNE works for newer models (e.g., LLaMA3-8B, Qwen3-8B).

4. The writing can be improved. The citation style is not correct.

**Questions:**

1. Will TUNE adapt to non-classification tasks (e.g., text summarization)? How would frequency decomposition handle sequential generation’s long-range dependencies?

2. What is the basis for hyperparameters like γ=2.0 and λ=0.7? Do they need adjustment for different model sizes or task types, and how sensitive is TUNE to their variation?

3. How does TUNE perform with longer task sequences (50+ tasks)?

---

### Note · Authors · 2025-11-14

I have read and agree with the venue's withdrawal policy on behalf of myself and my co-authors.